# Computer-Based Intervention Closes Learning Gap in Maths Accumulated in Remote Learning

**DOI:** 10.3390/jintelligence10030058

**Published:** 2022-08-18

**Authors:** Réka Ökördi, Gyöngyvér Molnár

**Affiliations:** 1Doctoral School of Education, University of Szeged, 32-34 Petőfi sgt., 6722 Szeged, Hungary; 2MTA-SZTE Digital Learning Technologies Research Group, Institute of Education, University of Szeged, 32-34 Petőfi sgt., 6722 Szeged, Hungary

**Keywords:** computer-based, mathematical reasoning, primary school, remedial education, game-based

## Abstract

Remote learning has reduced the mathematical performance of students. Mathematical reasoning is the critical skill that enables students to make use of all other mathematical skills. The aim of the present study was (1) to develop the mathematical reasoning skills of underachieving students and (2) to explore the application options, benefits and limitations of an online game-based intervention programme among third- and fourth-grade pupils (aged 9–11, N = 810). The content of the programme was designed to strengthen their basic maths skills in line with the curriculum. Beyond assigning the tasks, the digital intervention programme also performed the motivational, differentiation- and feedback-related tasks of the teacher. The reliability indices for pre-, post and follow-up test results proved to be high (Cronbach’s alpha = .90, .91 and .92, respectively). The effect size of the programme proved to be significant in both grades (d = .22 and .38, respectively). The results confirm the potential of the intervention programme to close, or at least significantly reduce the Covid learning gap in basic maths skills, without the need for additional teacher work—which is an important aspect of successful implementation—in areas which are the most challenging for 9-to-11-year-old pupils in the domain of mathematics.

## 1. Introduction

In the past three school years, 91% of school-aged children have been affected worldwide by short- or long-term remote learning or even repeated transition to digital education ([38]). Mathematics is one of the areas where students’ learning gap proved to be the highest due to remote learning and where the learning loss rate is comparable to the impact of the summer setback on the students’ achievement gap ([13]). That is, students learned little or nothing from the mathematics curriculum through remote learning during the pandemic.

Learning loss due to school closures has aggravated and enlarged inequalities in education ([3]; [13]; [15]; [39]). Children with low socio-economic status had more difficulties gaining access to devices, to the Internet and, consequently, to learning materials and online classes. They spent less time studying than they had before, since they lacked sufficient support from their parents.

Researchers reported three to five months of learning loss in maths at the beginning of the 2020–2021 school year ([12]; [13]; [23]). While school learning tends to reduce socio-economic disadvantages ([3]), remote learning has obviously enlarged differences, especially in mathematics, among primary students with social, mental or learning difficulties ([13]; [15]). These students had been lagging far behind their peers even during “normal” teaching before the pandemic, but their knowledge gap has been further exacerbated by school closures ([3]). Without targeted help and extra remedial lessons, the extent of students’ falling behind will not decrease ([39]) in the so-called Covid cohort.

The longer schools were kept closed, the greater the decline in students’ performance and the larger their knowledge gap, especially in mathematics. In Europe, schools were closed for a longer period in countries where inequalities in access to education are significant ([3]). One of these countries is Hungary, where the average level of knowledge and performance among second to eighth graders also proved to be lower at the beginning of the 2020–2021 academic year than in the same period during the previous two academic years. Similar trends have been published for the overall performance of Flemish primary students ([23]), for Dutch students ([13]) and for US students in mathematics ([19]). In the present paper we introduce a technology-based intervention programme for 9–11-year-old pupils and present the results of a large-scale study. The short-term aim of the intervention programme was to close or at least minimalize the COVID-19 learning gap of third- and fourth-grade pupils in the field of mathematics by developing their mathematical reasoning. More particularly, the long-term objective was to create a programme that could be used in school education that helps struggling students boost their understanding of the concept of multiplication and division, as well as improve their basic mathematical skills. The game-based programme consisted of 15 intervention sessions aligning with curriculum requirements and including both instructions and feedback to facilitate students’ work and substitute teachers’ remedial work. The computer-based intervention programme provided individual learning paths for students based on their performance. To the best of our knowledge, no online, empirically proven personalised intervention programme exists in mathematics, providing full support for pupils—without their teacher—so that they can improve their basic skills of mathematics in Hungary. Nevertheless, research shows that curriculum-appropriate use of virtual learning platforms and the immediate feedback they provide positively affect the mathematical performance of primary school pupils ([18]). Children supported with digital learning materials throughout the school year also perform significantly better on standardised maths assessments ([40]).

## 2. Aspects of Designing an Effective Computer-Based Intervention Programme for Students

In remote learning during the COVID-19 pandemic, there were significant disparities in students’ access to education. One of the major challenges in classroom teaching comes from the differences between students in terms of learning loss or stagnation following school closures. Intervention programmes are called for which are applicable both in the classroom and in home assignments and which can further reduce disadvantages, develop skills and aid students in catching up with their studies. In contrast to face-to-face or mostly frontal teaching, technology-based programmes can make the learning environment more motivating, offer personalised tasks and provide feedback to monitor progress ([9]; [27]). The use of technology in the teaching and learning process has the potential to personalise education and adjust its processes to the individual needs of the students, while minimising the invested effort of the teachers ([27]).

When developing and introducing a digital intervention programme in schools, it is essential to consider the preferences of the teachers who would implement the new learning resources in their classrooms ([6]; [11]; [14]; [33]). Teachers need appropriate content, as well as pedagogical and technological knowledge, to be able to successfully integrate the use of digital resources into the classroom ([26]). Appropriate technical tools, technical assistance and user guides for digital content should be available ([14]; [16]). In addition to these criteria, teachers are more willing to use new materials that fit into the crowded compulsory curriculum and can easily be aligned to the syllabus ([11]; [36]). Digital resources should support their own teaching style and practice ([11]). Teachers prefer learning tools with which learning outcomes can be measured and viewed.

Another feature of the digital intervention programme is that it allows for personalised individual learning. Whole-group instruction determines teaching practice, and, in many cases, this hinders the introduction of personalised IT tools in the classroom ([14]; [21]). Nevertheless, a particular potential advantage of digital learning resources is that they can be used to personalise the tasks assigned to students in line with the compulsory curriculum, based on students’ logged responses ([6]). This aids in differentiating and developing students who have fallen behind. Logging the responses provides feedback for both teachers and parents.

In addition to the previous criteria, when designing a computer-assisted intervention programme to foster successful learning among struggling students, several aspects must be considered ([34]). The technological tools needed to access digital learning materials must be available and suitable for all students. The content of the digital curriculum should be age appropriate as well as cognitively and pedagogically relevant to the needs of each student ([27]; [16]). Instructions must be divided into small segments. Students should be given the opportunity to practise the prerequisite skills before classroom instruction and the targeted skills afterwards to complete a task.

In addition to the previously noted student-related factors, students in the lower grades or students with reading difficulties should be given instructions by pre-recorded voice on the digital platform so that their reading performance does not affect their mathematical performance ([27]). Visual aids, such as animation, graphics and representations, should support their understanding of the material, help focus their attention and maintain their motivation ([34]). First-generation tasks like those available on paper and in textbooks ([28]), mixed with second-generation tasks, including multimedia ([32]), enhance student motivation in the 21st century ([10]). In primary school, content should be presented in a clean and simple interface with only those pieces of information and images which do not distract pupils ([24]; [34]). Students given continuous feedback boost their motivation and raise their limits of endurance.

## 3. Aim and Research Questions

The objective of this study is twofold. First, to develop a game-based, personalised intervention programme for third- and fourth-grade pupils in mathematics to develop their mathematical reasoning as well as to close the learning gap accumulated in the first two years of schooling during remote learning regarding basic maths skills. Second, to conduct a quasi-experimental research project to test the effect size of the intervention immediately after the intervention and three months later in a follow-up test with respect to different groups of pupils, that is, pupils with different levels of basic maths skills.

We thus aim to answer six research questions: (RQ1) How effectively can a computer-based intervention programme in mathematical reasoning develop basic maths skills among pupils aged 9 to 11 in the short and longer term (i.e., assessed in post- and follow-up tests)? (RQ2) How does the intervention programme impact on the mathematical skills of third and fourth graders with different educational backgrounds and experience in the short and longer term? (RQ3) What changes can be observed in the distribution of pupils’ performances in the pre-, post- and follow-up tests? (RQ4) What are the short- and long-term impacts on pupils’ thinking skills needed to understand the concept of multiplication and division and to solve specific word problems? (RQ5) Which starting level of mathematical reasoning is the most sensitive to the intervention programme in the short and longer term? (RQ6) How generalizable are the results on skill and sub-skills level? Are the effects proven by the project confirmed by latent level analyses using a latent change model in the intervention group and a no-change model in the control group?

## 4. Methods

### 4.1. Participants

The sample of the study consisted of third- and fourth-grade primary school pupils (aged 9 to 11). A total of 2187 pupils took part in the quasi-experimental study. After data cleaning, data from 810 pupils were included in the analyses. We have excluded pupils who had more than 50% missing data on one of the tests; based on time-on-tasks data, we also excluded pupils who just went through the tasks without spending a minimum amount of time needed at least to read or listen to the instructions; we have also excluded those intervention group pupils who completed less than half of the intervention programme. Finally, we only processed data from pupils who completed not only the pre-test and post-test, but also the follow-up test. After data cleaning, we applied propensity score matching: each pupil in the intervention group was matched with one pupil in the control group based on their grade (3 or 4) and pre-test performance. After data elimination and propensity score matching, data from 207 intervention–control group pairs in Grade 3 and 198 pairs in Grade 4 (a total of N = 810) was included in the analysis (Table 1).

### 4.2. Instruments

A computer-based multiplication and division test was used as pre-, post- and follow-up test to measure the effectiveness of the intervention in basic maths skills. The instructions were provided in both written and spoken formats. The test, comprising 35 items in total, consisted of three sub-tests: the multiplication subtest (15 items), the division sub-test (12 items) and the matching operations and word problems sub-test (8 items). When developing the multiplication sub-test, we took into consideration the one-to-many correspondence as the origin of multiplicative reasoning ([30]; [31]), but since children often first encounter multiplicative reasoning through repeated addition in textbooks, the sub-test involved exercises corresponding to both interpretations of multiplication. Similarly, on the division sub-test, the exercises are based on either of the two accepted interpretations. In tasks based on the partitive (sharing or grouping) model, we divided an amount into a given number of parts. In tasks based on the measurement (quotative, or repeated subtraction) model, we divided an amount into parts of a given size. Finally, for matching operations and word problems, the tasks included mathematical problems both in multiplication and division. Please note, that in this part, the questions required only choosing the right operations for the word problem, not performing them. While the first two sub-tests also assessed pupils’ numeracy skills, this sub-test focused exclusively on assessing their thinking skills. Therefore, children were not expected to perform the chosen operations; therefore, errors in calculation did not affect their performance of the task (Figure 1).

An online intervention programme was developed to boost individual achievement and promote individual progress, thus providing a means for differentiation in the classroom. The novelty of the intervention programme, which was aligned with the curriculum requirements, lies in the fact that it embeds the developmental process in a digital environment, as well as including instructions for the tasks and providing automatic feedback on pupil performance.

The programme consisted of 306 mathematical reasoning tasks, divided into fifteen intervention sessions. Each session offered individual learning paths, since based on pupils’ responses, the programme directed them further to an appropriate branch of tasks. In the case of an incorrect answer, either the pupil was aided in approaching the problem from a different perspective or additional visual and/or auditory assistance was provided. In another type of branch, we assumed a skill or vocabulary deficiency behind an incorrect answer, and, therefore, the programme offered specific tasks to pupils to improve those skills. In addition to assigning the tasks, the programme also comprised motivational and feedback-related elements. The tasks and further assistance provided also focused on developing thinking skills. The aim throughout was to help pupils understand the concepts of multiplication and division and to apply these operations in a meaningful way as the appropriate calculating procedures in different problem situations. (Figure 2) Instructions and questions were recorded, and pupils could listen to them as many times as they wished to overcome reading comprehension difficulties and to reduce the cognitive load.

One salient feature of the intervention programme is its use of so-called eDia activities (note that eDia is the name of the online platform on which the intervention programme was developed). One of the reasons for using eDia activities is that out of all subjects, grades earned in mathematics and reading and writing show the strongest correlation with using elements of visual communication ([35]). An element of visual communication carries a message which may enrich the recipient’s understanding either directly or indirectly. Therefore, in all the eDia activities of the programme, images and short videos (moving images) helped develop the concepts of multiplication and division. Students observed and/or created images, making visual communication a helpful tool for understanding maths problems (Figure 3 and Figure 4).

### 4.3. Design

The study followed a quasi-experimental design. Both the intervention and the control groups were assessed before (T1) and after (T2) the intervention programme. To assess the permanence of their development, pupils also completed a follow-up (T3) test three months after the end of the intervention programme (Figure 5). As a result of a propensity-score matching, pupils were assigned to the intervention or control groups according to their grades and pre-test scores. First, each child was matched with a classmate based on their pre-test results, then one of them was randomly assigned to the intervention and the other to the control group. Thus, the achievement distribution and the school-related factors (e.g., effect of the teacher) beyond the intervention programme were both balanced out.

The computer-based intervention programme and the pre-, post- and follow-up tests were administered via the eDia online platform ([8]; [27]). School or personal infrastructure is essential for the successful integration of technology-based resources into teaching maths because access to the pre- and post-tests as well as the intervention programme should be provided for. Tasks running on the eDia platform are accessible on all kinds of technological devices with an internet connection and a general browser, that is, on desktop computers, laptops, tablets or even mobile phones ([29]). Although the eDia platform was accessible outside school, data collection took place exclusively in the ICT rooms or classrooms of the participating schools.

Participating teachers received a short online briefing session—one hour in total—before the start and were provided with continuous help from the researchers while the programme was being implemented. The aim of the on-going assistance was to supply information about the concept, the platform and the process of the intervention. The detailed information provided to teachers in writing and the briefing included the following information: (1) how learning loss due to the lockdowns may affect primary school students’ mathematical skills; (2) what kinds of reasoning skills and curriculum content are targeted by the intervention; (3) what the structure of the intervention looks like; (4) how much time should be allocated to implementing the intervention programme; (5) a detailed description of the content, skills focus and task types for each session; and (6) how the eDia online platform can be used. During the training session, we agreed on the exact role of the teachers during the intervention. Teachers were given the following instructions: (a) pupils should be given no more than three sessions a week and one session per day, (b) one day off between two sessions would be preferable, and (c) pupils should complete at least eleven of the fifteen sessions. Teachers were asked to resolve any technical issues but were not allowed to help pupils solve mathematical problems during the intervention sessions. Thus, the same conditions were ensured for the intervention at each school.

The intervention lasted four to six weeks and took place in the school during regular school hours. Each session took approximately 10 to 20 min. Trained classroom teachers or afternoon teachers supervised the intervention. The eDia system logged, recorded and automatically scored the answers, and then administered the best fitting task to the pupil based on their previous answers. At some stages of the intervention session, eDia provided automatic positive or motivating feedback for the students (Figure 6 and Figure 7).

### 4.4. Procedures

To find developmental differences between the control and the intervention groups, we used standard statistical procedures such as the independent t-test and effect size (Cohen’s d). Latent mean differences could be interpreted as true differences in the measured construct; that is to test the generalisability of the results obtained on manifest level, the effect of the intervention programme was evaluated in the latent curve modelling framework, too.

We compared model fit indexes of CFI (comparative fit index), TLI (Tucker–Lewis index) with associated 90% confidence intervals and RMSEA (root mean square error of approximation) and the changes in fit indexes between the different models. We accepted CFI and TLI values > .90, RMSEA values < .08 (see [17]). The drop of CFI and RMSEA values was also used by identifying the best fitting model; if it was larger than .01, it supported the models on a different level ([5]).

Second, we assumed that this period of time is a sensitive period for developing pupils’ basic mathematics skills. Beyond manifest level, we tested this hypothesis on latent level and ran second-order linear latent growth model on sample level taking all three data collection points into account.

To test the developmental effect even more thoroughly, we further analysed the improvement between T1 and T2 and between T2 and T3 separately, both overall and by domains, for the intervention and control groups separately, based on the results obtained at the manifest level ([1]). First, we specified latent change model for both groups between T1 and T2, assuming that even normal school education without any extra intervention produce meaningful effects on pupils’ development. In this model, we estimated slope growth factor in both groups separately to capture any possible change. Second, we applied a latent change model for the intervention and a no-change model for the control group. Finally, we estimated a no-change model for both groups, assuming that neither normal school education nor the intervention has produced any meaningful effect.

For using the advantages of SGMs and to properly separate measurement error from reliable occasion-specific variance, at least two indicators per time point are required for the analyses. Therefore, based on the factor loading values and the procedure described by [37] ([37]) and [22] ([22]), we created test halves, that is two parcels for each time point. The composition of the parcels was identical for each of the three time points.

## 5. Results

The test–re-test consistency of the pre-, post-and follow-up tests and its subtests was good to excellent. The Cronbach-alpha varied between .90 and .92 (Table 2). The values of the reliability indices of each sub-test ranged from 0.70 to 0.86 (Table 2). That is, the test proved to be suitable for determining the developmental level of pupils’ multiplication and division skills.

### 5.1. Results for Research Question 1 (RQ1)

RQ1: How effectively can a computer-based intervention programme for mathematical reasoning develop basic maths skills among pupils aged 9 to 11 in the short and longer term (i.e., assessed in the post- and follow-up tests)?

No significant differences were found between the performances of the intervention and the control group (t = 1.2, *p* = .19) in the pre-test. Based on the post-test results both the intervention and control groups showed significant improvement on average (Table 3), however the intervention group significantly outperformed the control group (t = −8.08, *p* < .001). Pupils’ average achievement in the intervention group improved by one third of standard deviation as a result of school instruction and extra intervention. Based on the average development of pupils in the control group, half of this improvement was due to regular school instruction.

### 5.2. Results for Research Question 2 (RQ2)

RQ2: How does the intervention programme impact on the mathematical skills of third and fourth graders with different educational backgrounds and experience in the short and longer term?

Pupil performance by grade (3 or 4) was also analysed. The content of the intervention programme is the second-grade curriculum, so it was relevant to examine the impact of the intervention on pupils who had been exposed to this mathematical content in the previous school year and those who had encountered it for the first time two school years before. Furthermore, the school closures due to the pandemic affected different mathematical content in third or fourth grade.

There was a significant difference between the pre-test performance of the two grades (Table 4), i.e., the fourth graders started from a higher level of knowledge at the beginning of the intervention. The effect size was larger in the intervention group than in the control group in both grades immediately after the end of the intervention programme. The progress accelerated in both grades as a result of the intervention programme. In both grades, the effect sizes in the control and the intervention groups were the same or close to the same by the time of the follow-up test. The progress in third grade accelerated in both groups. In fourth grade, the intervention group’s progress slowed down slightly without the intervention, while the size of progress of the control group remained the same as in the post-test.

### 5.3. Results for Research Question 3 (RQ3)

RQ3: What changes can be observed in the distribution of pupils’ performances in the pre-, post- and follow-up tests?

The distribution curves (Figure 8) also show the improvement of both groups’ performances at each test date. However, the distribution curve for the follow-up test is shifted more to the right for the intervention group. In their case, the curve flattens out at lower achievement levels and peaks just at the high achievement values.

### 5.4. Results for Research Question 4 (RQ4)

RQ4: What are the short and long-term impacts on pupils’ thinking skills needed to understand the concept of multiplication and division and to solve specific word problems?

The pupils’ progress in all three domains was accelerated as a result of the intervention programme (Table 5). During the same period of time, the control group developed less their maths performance as a result of schooling alone. However, there is a difference between the two groups in the extent of progress assessed at the post-test in all three domains: the difference between the effect sizes was maintained only in the multiplication domain during the follow-up test. The effect sizes for the subtests of division and word problem were found to be almost identical in the follow-up test at the end of the school year.

### 5.5. Results for Research Question 5 (RQ5)

RQ5: Which starting level of mathematical reasoning is the most sensitive to the intervention programme in the short and longer term?

To examine the change according to performance levels, pupils were divided into three groups based on their pre-test scores. Group A consisted of pupils who scored more than one standard deviation lower than mean achiever; group B was formed out of the mean achievers; while high achiever, whose achievement proved to be significantly higher than that of the mean achiever belonged to group C.

In the post-test taken immediately at the end of the intervention, the rate of progress—in terms of [7]’s ([7]) convention for describing the magnitude of effect size—proved to be higher in the intervention groups than in the control groups at ability levels A and B (Table 6). Although the rate of improvement for group A was significantly reduced by the time of the follow-up test, the intervention group’s improvement was still greater than that of the control group. The effect sizes for level B performers in both the control and the intervention groups were found to be almost identical in the follow-up test at the end of the school year. For the best performing group C, the rate of improvement can be shown by comparing the pre-test and the follow-up test. The control group did not make significant progress in the assessed mathematical areas, while the intervention group made significant progress by the end of the school year.

### 5.6. Results for Research Question 6 (RQ 6)

RQ6: How generalizable are the results at skill and sub-skills level? Are the effects proven by the project confirmed by latent level analyses using a latent change model in the intervention group and a no-change model in the control group?

First, we tested a measurement model with all indicators combined under one general factor then we tested a three-dimensional model assuming that it is adequate to separate mechanisms of multiplication, division and reasoning. The third-dimensional model fitted the data better (Table 7).

Second, we created two parallel three-dimensional forms of the test based on the factor loadings. Cronbach’s alphas were good (>.80), and correlations were .822, .836 and .861 at the three-time points, respectively. As hypothesized, the first model fitted the data the best (Table 8). In other words, the latent change model for both groups confirmed developmental changes in both the intervention and the control groups. We have detected remarkable changes, indicated by a significant increase in the slope factor (s = .506) in the intervention group compared to the control group (s = .244). The negative correlation of the intercept and slope values (r= −.339) indicated that pupils with lower ability level were more responsive to the intervention, they developed more than their peers as a result of the intervention.

Third, we monitored latent changes in the intervention and control groups at sub-test level (Table 9). In the case of multiplication, the latent change model fitted the data the best, that is, latent level analyses supported results obtained on manifest level (see Results regarding RQ4). There was a significant difference in the slope factor describing the speed and magnitude of the development (control group: s = .271, intervention group s = .462), confirming the accelerating power of the intervention. The intercepts of the latent curves were not significantly different (2.124), i.e., the ability level of both groups was the same at the time of the pre-test. The negative intercept—slope correlation suggested that those with lower ability levels improved faster their skills than those with higher ability levels, again confirming results at manifest level.

Similar results were found in the case of division, i.e., the latent change model fitted the data best, confirming developmental changes in both the intervention and the control groups. The difference in the slope factor (control group: s = .136, intervention group: s = .363) indicated significant differences in the speed of development in favour of the intervention group. As the intercepts of the latent curves were not significantly different (I = 1.974), i.e., the ability level of both groups was the same at the time of the pre-test, as a result of the intervention we managed to speed up the development of the pupils in the intervention group on average in comparison to the control group. The negative intercept—slope correlation suggested that pupils with lower ability levels improved their skills in division faster than those with higher ability levels as a result of the intervention, confirming results at manifest level.

In the case of word problems, the second model fitted the data the best, indicating no skill development for the members of the control group on the latent level and significant change in the skill level of the members of the intervention group on average (s = .309), which also confirmed the generalizability of the results obtained on manifest level.

## 6. Discussion

We have presented an evaluation study of a game-based mathematical reasoning intervention programme, which proved to be appropriate to close or at least significantly reduce the learning gap in basic maths skills among children between the ages of 9 and 11. The study followed a quasi-experimental design. The content of the intervention programme was built on curriculum requirements and on different conceptualizations of multiplication and division. The fifteen sessions consisted of 306 tasks, providing personalised learning paths for the pupils. As a by-product, we created the pre-, post and follow-up test, a reliable online instrument to assess the developmental level of pupils’ basic maths skills, which can be used effectively, even independently of the intervention programme.

With regard to RQ1, significant improvements were observed in all the pupils’ performances, in the control group as well as in the intervention group. This indicates that there is a sensitive period in children’s cognitive development, regardless of interventions other than school education. However, pupils who completed more than half of the online intervention sessions improved their skills by one third of standard deviation by the end of the intervention¸ while the progress of the control group was only half of the intervention group’s improvement.

As for RQ2, when looking at the effect size in the third grade, it can be concluded that without targeted intervention, overcoming accumulated deficiencies from the previous grade is extremely slow ([39]). Since the failure to catch up and the lack of adequate development of skills can slow down or even prevent the acquisition of further mathematical content, early intervention seems necessary for later academic success in mathematics ([4]). This can be confirmed by the fact that during the intervention period, third-grade pupils have been revising and extending their knowledge of multiplication and division acquired in second grade mathematics lessons, according to the National Core Curriculum ([25]). It is in this context that the difference between the improvement of the control and the intervention groups is one third of the standard deviation in favour of the latter.

In Grade 4, progress is stronger in both the control and intervention groups than in Grade 3. It can be assumed that fourth graders had been exposed to multiplication and division not only in second grade, but also in third grade, so they probably had had more practice by the time they started fourth grade. However, comparing the progress of the intervention group with that of the control group, we can see that the effect of the intervention programme is stronger even in this grade than that of schooling alone. This means that, targeted intervention in this grade is also essential so that pupils can overcome the difficulties. Moreover, the intervention programme is also effective for this age group, not just for one-year younger peers.

With regard to RQ3, the distribution curve of the intervention group for the follow-up test flattens out more at lower achievement levels and peaks more at the high achievement values than the distribution curve of the control group. An important piece of information for interpreting the results is that, according to the National Core Curriculum ([25]), children in both the third and fourth grades must have been doing multiplication and division in maths classes during the period of the intervention programme. In the third grade, the topic was a revision of the second -grade material, i.e., performing operations in the range of 0–100, and in the fourth grade, the topic was an extension of oral operations in a larger range of numbers.

Even though the difference between the rate of progress of the control and intervention groups between the post- and follow-up test is not significant, the trend is clear from the results. Despite the short duration and time span of the intervention programme, the development of the intervention group accelerated. The data shows that in the period between the post- and follow-up tests, without the explicit appearance of the topic in the curriculum, the intervention group members better understood and more reliably retained the concepts of these operations than the members of the control group.

As for RQ4, the pupils’ progress in all three areas was accelerated by only one or two 15–20 min long interventions per week. However, the difference between the effect sizes was maintained only in the multiplication domain during the follow-up test. This result reflects the design of the intervention programme. This programme of only 15 sessions had to upset the balance between three broad topics—multiplication, division and word problems—and, consequently, the greatest emphasis was placed on understanding the concept of multiplication. This emphasis is reflected in the pupils’ performances. Having said this, it is important to note that their progress accelerated in all three areas over the intervention period. Subsequently, they only maintained this rate of progress in multiplication, since they spent more time doing multiplication tasks during the intervention. In other words, the time invested in understanding the concept of basic operations pays off in the long run ([4]).

As far as RQ5 is concerned, an intervention programme of this nature is effective if it improves the performance of those lagging behind. As for skills and academic achievement, based on the performance-based groupings, we can conclude that the basic maths skills of those 9-to-11-year-old pupils who were struggling the most improved the most through the intervention. The rate of progress is higher in the intervention groups than in the control groups at ability levels A and B. The direction of the change in performance-based groupings is in line with our intention that the intervention programme should aim to improve the computational and thinking skills of the laggards. This result confirms earlier research findings that computer-based intervention programmes are suitable even for struggling students ([2]; [20]). Our results indicated that the newly developed online intervention programme for mathematical reasoning can be used effectively in primary school to improve students’ basic maths skills, both in multiplication and division and in identifying which operations of multiplication or division are needed to solve specific word problems.

As for RQ6, we found that at test level the latent change model fitted both the control and intervention groups. It should be noted that the targeted maths skills may show a normative developmental increase at this age. It is also observed that children at this age are receptive to developing these skills at school. The analysis also confirms that pupils with lower performance levels at pre-test developed their maths skills more than their better performing peers through the intervention programme we have developed. Further analyses at sub-test level also confirmed the results observed in the manifest approaches. In multiplication and division, the performances of pupils of both groups improved, but the development of the intervention group and overcoming their deficits from the previous school years was accelerated by the intervention programme. The word problem subtest, however, shows that only the intervention group made progress in this area, while the control group did not. In other words, the intervention programme can indeed promote conceptual understanding of multiplication and division beyond the routine practice of basic operations.

## 7. Conclusions

To sum up, the results suggest that this online intervention programme is successful. It develops 9-to-11-year-old pupils’ basic maths skills (in particular, multiplication and division), using a wide range of mathematical reasoning tasks and media. The findings indicate that both the mathematical skills and the subject knowledge expected of primary school students can be significantly and effectively developed in a computer-based personalised environment, even without the continuous presence of the teacher ([6]; [14]; [16]; [34]). An intervention programme has been created which is easily accessible both inside and outside school. The research shows that it is possible to compensate for the disadvantages of laggard learners in mathematics with an innovative online intervention programme that takes over all the teacher’s tasks and provides students with appropriate subject content and a personalised learning path.

## 8. Limitations and Further Studies

The limitations of the study follow from the large amount of data loss during the research. During the intervention period, many children irregularly attended school and there were short school closures even during the intervention period. Due to the rigorous methodology, only data from children who participated in the whole process were kept (pre-, post and follow-up tests, less than 50% missing data, more than the half of the intervention programme completed). As a result, data from only 810 children out of 2187 were included in the analysis.

A further limitation is that the intervention is explicitly based on the school curriculum, helping those who are lagging behind to catch up. Thus, the teaching style and methods chosen by teachers in each school may have had an impact on the outcomes of both groups.

As this intervention programme, developed by professionals, was found to be effective, further research could be conducted to have certain groups of pupils learn the curriculum through the programme, without any teacher intervention. In addition, it would be worthwhile to assess the effectiveness of the programme in higher grades.

## Figures and Tables

**Figure 1 jintelligence-10-00058-f001:**
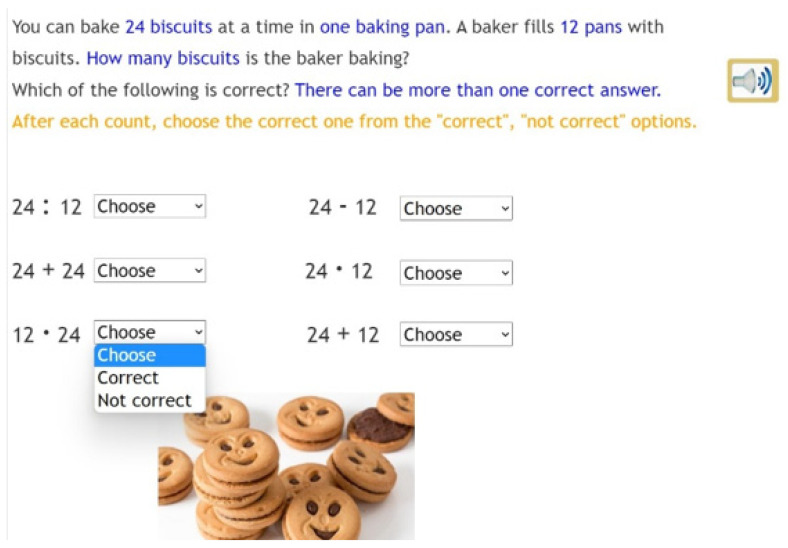
Example task from the word problem sub-test.

**Figure 2 jintelligence-10-00058-f002:**
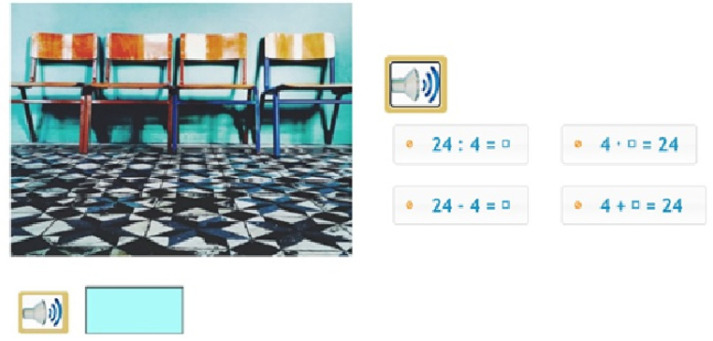
Example of a task (Transcripts: Step 1: This is how many chairs are placed next to each classroom door in the school corridor. How many chairs are there next to each classroom door? Step 2: How many classrooms are there on the first floor if there are 24 chairs for students? Step 3: (in case of a wrong answer): Let’s think together. There are four chairs next to each classroom door. That is, there are four chairs next to the classroom door of class A, there are four chairs next to the classroom door of class B, and so on. How would you calculate how many classrooms there are if there are eight chairs in the corridor? Once you have figured it out, calculate how many classrooms there are if there are 24 chairs in the corridor.; Step 4: (in case of a repeated wrong answer) Children are required to fill in a table on the screen with the number of chairs for classroom 1, 2, 3 etc. Thereafter, they should answer the following question: Which operation could you use next time to calculate the number of chairs more easily? (See right side of Figure 2).

**Figure 3 jintelligence-10-00058-f003:**
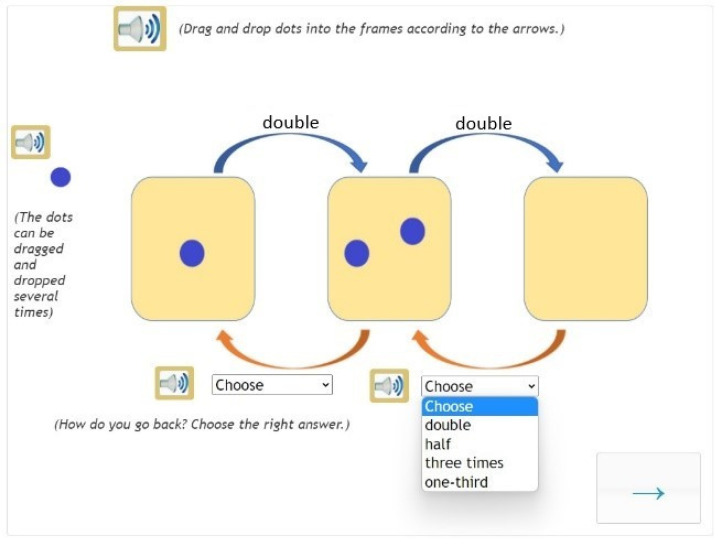
Example of a multiplication task.

**Figure 4 jintelligence-10-00058-f004:**
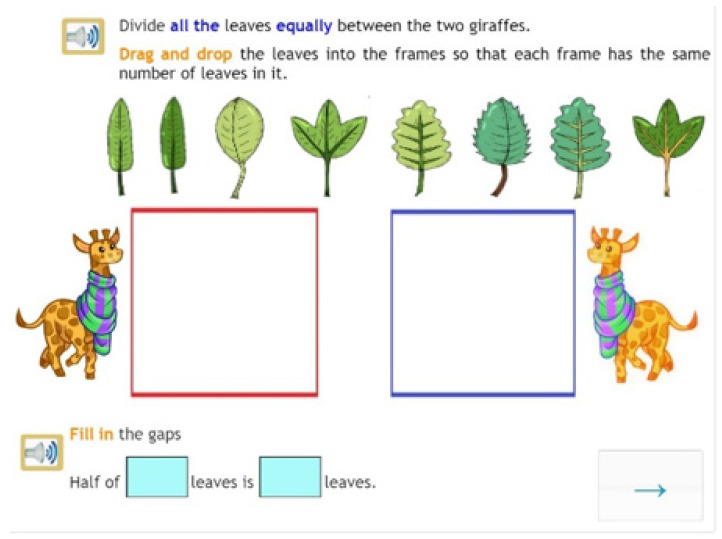
Example of a complex task.

**Figure 5 jintelligence-10-00058-f005:**
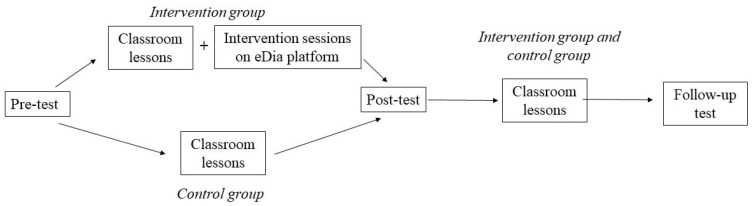
Educational processes in the intervention group and in the control group.

**Figure 6 jintelligence-10-00058-f006:**
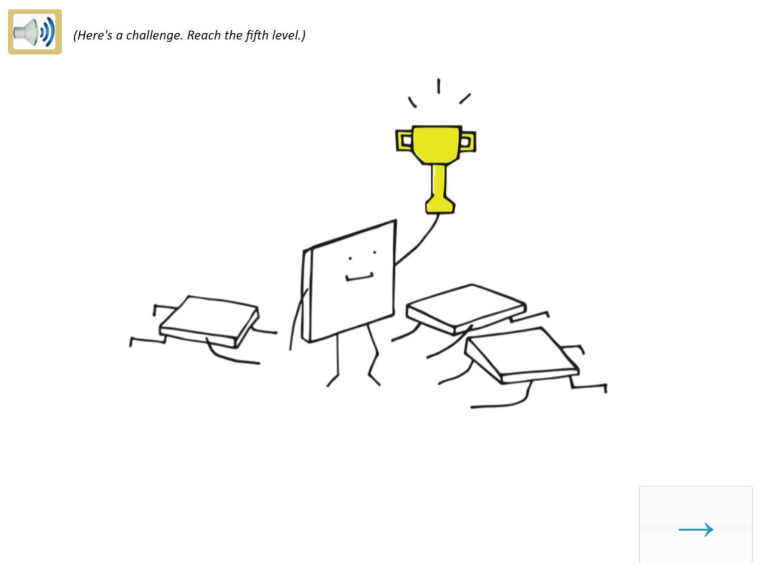
Example of a motivating feedback.

**Figure 7 jintelligence-10-00058-f007:**
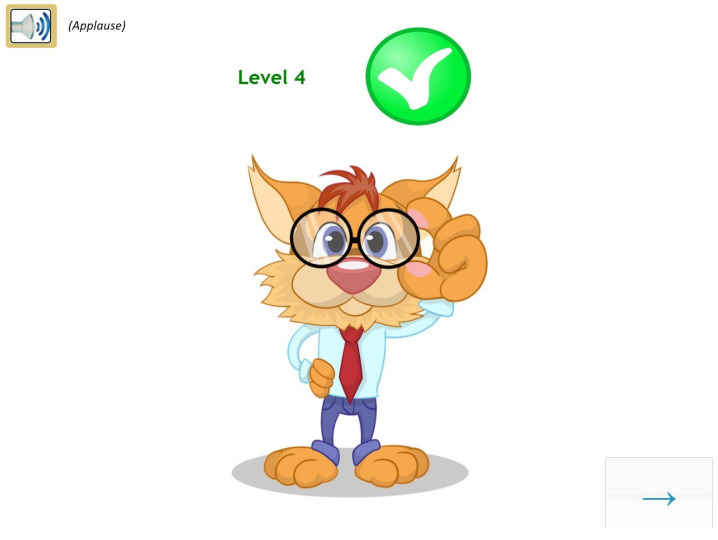
Example of a positive feedback.

**Figure 8 jintelligence-10-00058-f008:**
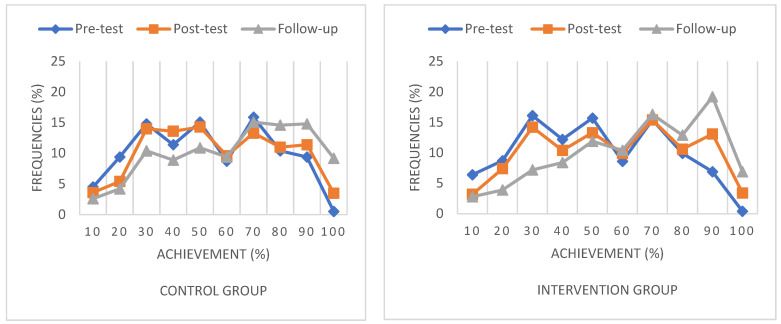
Distribution curves of control and intervention groups in the pre-, post- and follow-up tests.

**Table 1 jintelligence-10-00058-t001:** Sample of the study.

Grade	N of Participants	N_intervention Group	N_control Group
3	414	207	207
4	396	198	198

**Table 2 jintelligence-10-00058-t002:** Reliability indices for pre- post- and follow-up test results.

	Test	Multiplication (15 Items)	Division (12 Items)	Word Problem (8 Items)
Pre-test	0.90	.80	.84	.70
Post-test	0.91	.82	.86	.71
Follow-up	0.92	.86	.84	.75

**Table 3 jintelligence-10-00058-t003:** Within group differences on the pre-test, post-test and follow-up test.

	N	Pre-Test (T1)		Post-Test (T2)		Follow-Up (T3)		t (T1_T2)	*p* (T1_T2)	d_T1_T2	d_T2_T3
		M	SD	M	SD	M	SD				
Intervention	405	50.2	23.4	57.1	24.4	64.4	24.0	−8.08	<.001	0.29	0.30
Control	405	52.3	23.6	56.0	24.0	63.2	24.8	−4.04	<.001	0.15	0.29

**Table 4 jintelligence-10-00058-t004:** Change in performances according to performance-based groupings (in terms of Cohen’s-d) in pre-, post- and follow-up test.

	N	Pre-Test (T1)		Post-Test (T2)		Follow-Up (T3)		t T1_T2	*p* T1_T2	d_T1_T2	d_T2_T3
		M	SD	M	SD	M	SD				
Intervention Grade 3	207	47.2	22.9	52.2	24.2	60.9	25.3	−4.0	<.001	0.22	0.35
Control Grade 3	207	50.1	24.1	52.1	23.7	60.8	25.8	−7.6	<.05	0.08	0.35
Intervention Grade 4	198	53.3	23.5	62.2	23.5	68.1	21.9	−7.8	<.001	0.38	0.26
Control Grade 4	198	54.8	22.9	60.1	23.6	65.7	23.7	−4.0	<.001	0.23	0.24

Note. d: Cohen-d.

**Table 5 jintelligence-10-00058-t005:** Change in domain-related performances according to post and follow-up tests’ results (in terms of Cohen’s-d).

	d_T1_T2		d_T2_T3	
	Intervention	Control	Intervention	Control
Multiplication subtest	0.31	0.22	0.31	0.24
Division subtest	0.19	0.09	0.25	0.27
Word problem subtest	0.26	0.05	0.22	0.25

Note. T1: pre-test, T2: post-test, T3: Follow-up, d: Cohen-d.

**Table 6 jintelligence-10-00058-t006:** Change in performances according to performance-based groupings (in terms of Cohen’s-d) in pre-, post- and follow-up test.

	N	Pre-Test (T1)		Post-Test (T2)		Follow-Up (T3)		t	*p*	d_T1_T2	d_T2_T3
		M	SD	M	SD	M	SD				
Intervention_A	72	16.6	7.5	32.9	15.8	40.5	19.5	−8.4	<0.001	1.32	0.42
Control_A	68	18.1	6.1	33.3	17.6	38.3	20.9	−7.4	<0.001	1.15	0.25
Intervention_B	250	49.8	15.8	56.1	21.2	64.4	21.3	−6.5	<0.001	0.34	0.39
Control_B	243	49.8	13.6	53.5	20.4	61.5	21.4	−3.1	<0.001	0.21	0.38
Intervention_C	83	83.9	6.6	82.0	13.9	86.9	12.0	−2.2	<0.05	-	0.31 (T1_T3)
Control_C	94	84.4	6.3	79.1	15.8	85.3	14.5	5.8	n.s.	-	0.08 (T1_T3)

Note: A: low achiever on the pre-test (pupils who scored more than one standard deviation lower than mean achiever on the pre-test), B: average achiever on the pre-test, C: high achiever on the pre-test (pupils who scored one standard deviation higher than average achiever on the pre-test).

**Table 7 jintelligence-10-00058-t007:** Goodness of fit indices for testing dimensionality of the construct under investigation.

Model	Χ2	df	CFI	TLI	RMSEA	CI
Three-dimensional	2742.9	556	.924	.924	.070	[.067 .075]
One-dimensional	3230.0	464	.902	.896	.086	[.083 .089]

Note. df = degrees of freedom; CFI = comparative fit index; TLI = Tucker–Lewis index; RMSEA = root mean square error of approximation; CI = confidence interval for RMSEA; χ2 and df are estimated by WLSMV.

**Table 8 jintelligence-10-00058-t008:** Goodness-of-fit indices for the tested models on test level.

Model	χ2	df	CFI	TLI	RMSEA [90% CI]
Latent change model for both groups	171.3	10	.935	.922	.200 [.174 .227]
No-change model for the control and latent change model for the intervention group	233.3	11	.910	.902	.224 [.199 .249]
No-change model for both groups	250.8	12	.903	.903	.222 [.199 .246]

Note: CFI, comparative fit index; TLI, Tucker–Lewis index; RMSEA, root mean square error of approximation; CI, confidence intervals.

**Table 9 jintelligence-10-00058-t009:** Goodness-of-fit indices for the tested models on sub-test level.

	Model	χ2	df	CFI	TLI	RMSEA [90% CI]
Multiplication	Latent change model for both groups	73.6	10	.960	.951	.125 [.100 .153]
	No-change model for the control and latent change model for the intervention group	91.7	11	.949	.944	.135 [.110 .161]
	No-change model for both groups	136.4	12	921	921	.160 [.137 .185]
Division	Latent change model for both groups	107.9	10	.928	.913	.156 [.130 .183]
	No-change model for the control and latent change model for the intervention group	112.2	11	.925	.919	.151 [.126 .177]
	No-change model for both groups	134.6	12	.910	.910	.159 [.135 .184]
Word problem	Latent change model for both groups	12.4	10	.996	.996	.025 [.000 .062]
	No-change model for the control and latent change model for the intervention group	12.4	11	.998	.998	.018 [.000 .057]
	No-change model for both groups	29.9	12	.974	.974	.061 [.034 .089]

Note: CFI, comparative fit index; TLI, Tucker–Lewis index; RMSEA, root mean square error of approximation; CI, confidence intervals.

## Data Availability

Data are available upon request due to privacy restrictions.

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
