# Peer review of "Computer-Based Intervention Closes Learning Gap in Maths Accumulated in Remote Learning"

_jintelligence, 2022, doi:10.3390/jintelligence10030058_

Round 1

Reviewer 1 Report

The manuscript has interesting research setting and proper theoretical background. The article is well-presented with clear structures, easy-to-understand languages, and enough descriptions of the research. However, some concerns need to be clarified to improve further versions of this manuscript.

1. The reliability and validity of the computer-based multiplication and division test should be provided.

2. Are there urban-rural differences in participants during the computer-based intervention?

3. The text in section “Design” and “Procedures” is quite dense. It would be good to try to generate some diagram that could give a more synthetic view of the topic.

4. It doesn't seem to be explained clearly. How did the control group learn? Did the control group use the computer (eDia platform) to learn?? What is the main difference in learning methods between the control group and the experimental group? What is the definition of the short term and longer term?  It is necessary to give more details on the methods and experiment itself as well as the introductions of the eDia online platform.

5. Figure 5. Be consistent in the use of words. (intervention groups or experimental group)

6. Check the reference format is correct such as “Engzell et al. 2021” or “Capone et al., 2021” ?

Reviewer 2 Report

Well structured paper, logical follow up of research questions, clear reasoning on the data found in research.

Reviewer 3 Report

The article reacts to the recent problems in teaching mathematics - the learning gap at underachieving pupils due to online teaching. The designed intevention programme was tested in practice and was the source for six research questions which were thoroughly discussed.

I suggest to use the term "pupil" throughout the whole article instead of "student" which should be used for tertiary education.

In lines 67-79  a different size of leters is used - was it taken from a different source?

Reviewer 4 Report

This article reports a very interesting and relevant study. It is well written, and readability is quite good. The results presented are relevant for both research and teaching practice.

There are minor aspects that could be addressed to further improve the article.

The text size in the second half of the last paragraph of section 1 is smaller than in the rest of the manuscript.

The last sentence of this paragraph begs the question of whether there are studies, with the characteristics described, that show improvements in students’ basic skills of mathematics in countries other than Hungary.

The sentences in lines 105-110 should probably be backed up by some appropriate references.

There is an apparent contradiction between what is said in lines 136-138 and in lines 142-144.

“3 and 4 grade students” should probably be replaced by “3rd and 4th grade students” or by “students in grades 3 and 4”.

In figure 3 there is a word that is not translated to English (“kétszerese”).

It would be relevant to know the duration of the “short online training session” mentioned in line 308.

It would be interesting to have some examples of the “automatic positive or motivating feedback for the students” (line 335).

In line 396, t=-10,1 seems different from the value given in table 3 (-8.08).

In the middle of line 562, a full stop is missing. In line 681, a space is missing before the parenthesis.

Finally, there are 7 self-citations in a total of 38 references (over 18% of total references). The authors should consider whether they are all really necessary.

Round 2

Reviewer 1 Report

This paper has been revised in accordance with the comments of the reviewers. Therefore, I think it is acceptable for publication.

Reviewer 4 Report

All the aspects identified for improvement have been conveniently addressed, so I believe that the manuscript has been sufficiently improved to warrant publication in the Journal of Intelligence.